# The Potential of UAV Imagery for the Detection of Rapid Permafrost Degradation: Assessing the Impacts on Critical Arctic Infrastructure

**Soraya Kaiser** [1,2,*], **Julia Boike** [1,2], **Guido Grosse** [1,3] **and Moritz Langer** [1,2]

1 Permafrost Research Section, Alfred Wegener Institute Helmholtz Centre for Polar and Marine Research, Telegrafenberg A45, 14473 Potsdam, Germany

2 Geography Department, Humboldt-Universität zu Berlin, Unter den Linden 6, 10099 Berlin, Germany

3 Institute of Geosciences, University of Potsdam, Karl-Liebknecht-Str. 24-25, 14476 Potsdam, Germany

* Correspondence: soraya.kaiser@awi.de; Tel.: +49-331-58174-5447

**Abstract:** Ground subsidence and erosion processes caused by permafrost thaw pose a high risk to infrastructure in the Arctic. Climate warming is increasingly accelerating the thawing of permafrost, emphasizing the need for thorough monitoring to detect damages and hazards at an early stage. The use of unoccupied aerial vehicles (UAVs) allows a fast and uncomplicated analysis of sub-meter changes across larger areas compared to manual surveys in the field. In our study, we investigated the potential of photogrammetry products derived from imagery acquired with off-the-shelf UAVs in order to provide a low-cost assessment of the risks of permafrost degradation along critical infrastructure. We tested a minimal drone setup without ground control points to derive high-resolution 3D point clouds via structure from motion (SfM) at a site affected by thermal erosion along the Dalton Highway on the North Slope of Alaska. For the sub-meter change analysis, we used a multiscale point cloud comparison which we improved by applying (i) denoising filters and (ii) alignment procedures to correct for horizontal and vertical offsets. Our results show a successful reduction in outliers and a thorough correction of the horizontal and vertical point cloud offset by a factor of 6 and 10, respectively. In a defined point cloud subset of an erosion feature, we derive a median land surface displacement of −0.35 m from 2018 to 2019. Projecting the development of the erosion feature, we observe an expansion to NNE, following the ice-wedge polygon network. With a land surface displacement of −0.35 m and an alignment root mean square error of 0.99 m, we find our workflow is best suitable for detecting and quantifying rapid land surface changes. For a future improvement of the workflow, we recommend using alternate flight patterns and an enhancement of the point cloud comparison algorithm.

**Keywords:** permafrost degradation; consumer-grade unoccupied aerial vehicle; North Slope Alaska; land surface displacement; point cloud alignment; structure from motion; M3C2

## 1. Introduction

The Arctic has experienced an average warming of more than twice as high as the global mean temperature over the past few decades [1]. This warming is causing widespread degradation of permafrost, reducing ground stability and causing substantial landscape changes (e.g., [2]). Numerous studies report an increase in landscape disturbances in permafrost regions with high ground ice content (e.g., [3,4]). The observed landscape dynamics are caused by, but not limited to, processes such as thaw slumping, landslides, thermokarst, and ground subsidence [2,5]. These landscape changes entail far-reaching ecosystem modifications affecting the water, heat, and nutrient balances as well as devastating consequences for infrastructure in Arctic communities (e.g., [6–8]). Models project 69% of Arctic infrastructure to be affected by near surface permafrost degradation by 2050, affecting approximately 3.6 million people [9]. Many Arctic communities report damages to their residential buildings, sewage, and water supply systems, two thirds of

them occurring since 2019 alone [10]. As most Arctic infrastructure is remote and affected by access limitations due to logistical and climatic challenges, instantaneous maintenance or restoration in case of a failure often is extremely challenging, potentially leaving flooded or damaged "lifelines" such as roads impassable for several weeks and even months. These data underline the necessity of reliable monitoring techniques of landscape-scale permafrost degradation in order to quantify risks for infrastructure and protect infrastructural elements. With the recent generation of low-cost and easy-to-operate unoccupied aerial vehicles (UAVs), an opportunity emerges to provide fast and cost-effective monitoring of 3D ground surface changes along infrastructural elements that is both sufficiently detailed as well as easy to operate by local citizens. The use of these UAVs, therefore, presents an asset to local environmental observers or emerging citizen science projects, such as the UndercoverEisAgenten (www.undercovereisagenten.org, accessed on 28 November 2022)—a project in which students in Canada and Germany work together to understand permafrost degradation processes by analysing image data acquired with a low-level UAV setup.

Several studies have already used UAVs to monitor land surface changes attributed to permafrost thaw. Successfully monitoring retrogressive thaw slumps [11,12], quantifying changes in sediment volume [13,14], and deriving land surface displacements at road embankments [14] show the potential of UAVs for thorough change detection at the centimeter-scale.

These studies, however, apply technically advanced real-time kinematics (RTK) drones or complementary differential GPS (dGPS) measurements for creating ground control points, as reliable spatial referencing for multi-temporal analysis is a known challenge in tundra landscapes due to unstable ground and limited features that can be used as permanent spatial reference points.

In this study, we therefore evaluate the use of an off-the-shelf, low-cost, non-RTK UAV without the application of ground control points for detecting local ground surface changes caused by permafrost degradation and determine the usefulness of this simplified approach to assess short-term risks to Arctic infrastructure. Ground control points are only used in a final quality assessment to demonstrate spatial accuracy. In particular, we apply Structure from Motion (SfM) for generating 3D point clouds from the 2D drone images to quantify sub-meter changes of an erosion feature close to the Dalton Highway, a key supply line that is essential to the Prudhoe Bay oil industry. Finally, we estimate future risks to the highway by deriving the magnitude and direction of the permafrost erosion process as observed with the UAV data. In order to design a UAV-based risk assessment workflow that would provide local environmental observers, communities, and citizen scientists with the needed tools at no extra costs, we follow the precondition to only use open-source and free software.

## 2. Materials and Methods

### 2.1. Study Area

The study site has an extent of 31.8 ha and is located at milepost (MP) 318 of the Dalton Highway on the North Slope of Alaska (see Figure 1). The landscape at the study site is characterized by lowland tundra mainly vegetated by tussock sedges and dwarf shrubs [15]. The study site is further characterized by ice-wedge polygons which are typical for this region and indicate ice-rich permafrost [16–18] (see area west of the Dalton Highway in Figure 1). The presence of thermo-erosion gullies indicates active permafrost degradation processes [17]. Based on ArcticDEM data [19], the study area consists of flat terrain with an average elevation of 392 m a.s.l. with a slightly elevated topography of up to 405 m at its western border. In the center of it, the Dalton Highway crosses from south to north and in the south, an access road leads from the highway to the bank of the Sagavanirktok River in the East. Parallel to the highway on the eastern side, a trench-like structure can be observed, which might mark the installation of a cable. The highway itself is a vital supply line for the Prudhoe Bay oil exploration sites in and around Deadhorse and is frequented by commercial traffic throughout the year. The northernmost exploration

site started developing after oil was discovered in 1968 [17] and is an important factor for Alaska's economy [20].

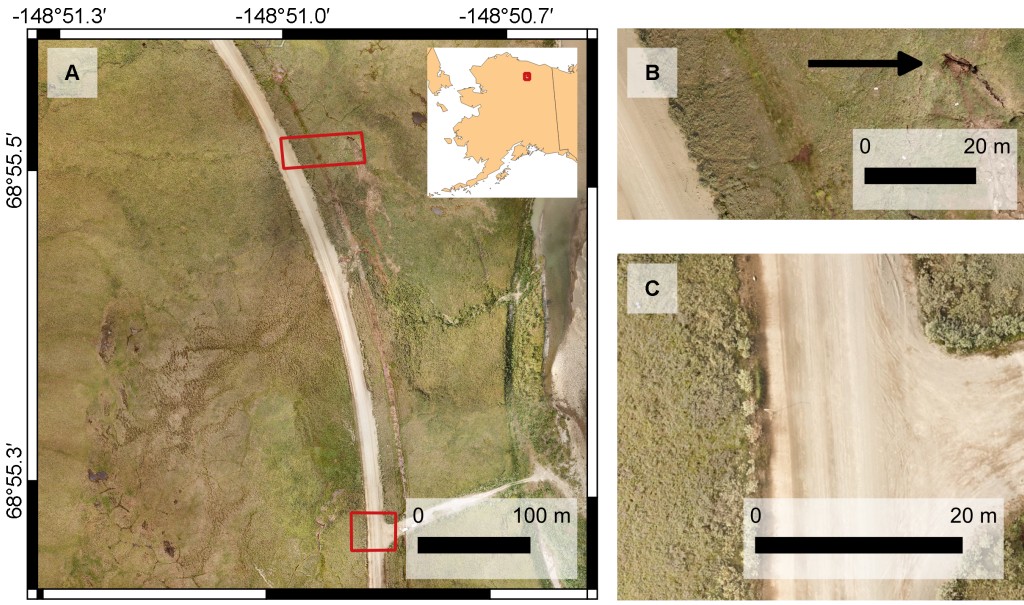

**Figure 1.** Map of the study area: (**A**) shows an orthophoto mosaic (RGB) of the study area along the Dalton Highway on the North Slope of Alaska. The subfigures on the right show the areas used for assessing the performance and accuracy of the workflow; (**B**) shows the area around an erosion feature assumed to undergo degradation; (**C**) depicts the road junction of the Dalton Highway, which is considered to be stable relative to the surrounding tundra. Raw images of the orthophoto mosaic were acquired on 18 July 2019, and stitched together using WebODM (see Section 2.3). The world border dataset for the inset map of Alaska in (**A**) is available from http://thematicmapping.org/ (accessed on 26 June 2021) and licensed under CC BY-SA 3.0 https://creativecommons.org/licenses/by-sa/3.0/ (accessed on 28 March 2022).

### 2.2. Image Acquisition

We acquired images in the consecutive summers of 2018 and 2019 using a DJI Mavic Pro quadrocopter over the study site along the Dalton Highway (Figure 1). The off-the-shelf UAV was equipped with a GPS and a gimbal [21] that tilts the camera and ensures stable image acquisition even in the face of wind turbulence. The camera captured RGB images at a resolution of 12 MP (4000 × 3000 pixels) [21]. We prepared and operated flight missions with the mobile drone flight application Drone Harmony [22]. To obtain images in nadir, we used the top-down mapping [23] acquisition plan and chose a height of 80 m above ground level (a.g.l.) with an overlap of 75% along-track and 65% across-track, respectively. Additionally, we operated a Point Of Interest (POI) flight plan [23] at a height of 60 m a.g.l. and an acquisition angle of 45°. In this acquisition mode, the drone obtains the images while circling around the erosion feature and rotating around itself to keep the feature in focus. Table 1 provides the exact dates and specifications of image acquisition.

**Table 1.** Acquisition date, number of images and ground sampling distance (GSD) of drone imagery, and spatial resolution of photogrammetry products.

| Acquisition Date | Number of Images | GSD [cm] | Orthophoto Mosaic [cm] | DSM [cm] |
|---|---|---|---|---|
| 7 August 2018 | 445 | 2.7 | 4 | 17 |
| 18 June 2019 | 522 | 2.7 | 4 | 18 |

*2.3. Photogrammetric Processing*

The processing of the acquired imagery entails the camera calibration with a subset of the RGB images and the photogrammetric processing to create orthophoto mosaics, digital elevation models, and 3D point clouds for each acquisition year. The point clouds are subsequently denoised and aligned to each other in order to compute distance parameters to each other, which provide information on the direction and magnitude of the erosion feature, see Figure 2.

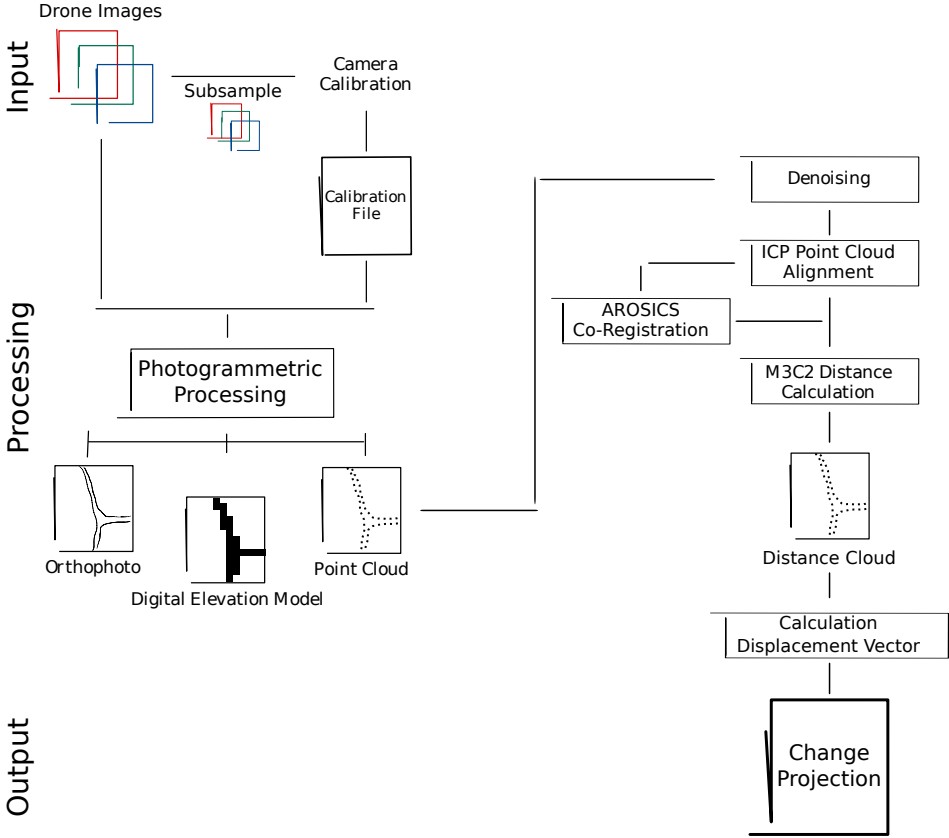

**Figure 2.** Workflow design for the detection and projection of land surface changes along critical infrastructure, subdivided into data input, photogrammetric and point cloud processing, and the resulting output.

2.3.1. Calibration

The photogrammetric processing of images, which were acquired with parallel, nadir flight paths often introduces systematic deformations to the final 3D products [24,25], known as "doming". To prevent these deformations, several studies suggest obtaining complementary oblique images of the study area [25,26], mitigating the occurrence of systematic deformation by 50% [27]. As battery capacity is limited, we chose a subset in the NE corner of the study area (see Figure 3) for a self-calibration of the lens to prevent a deformation effect.

This subset area was imaged in addition to the regular top-down mapping at an angle of 45° (POI flight, see Section 2.2) and a flying height of 60 m a.g.l. We processed these images with the default settings for high-resolution photogrammetry products in WebODM Lightning. The software generated a camera calibration file containing information on radial and tangential distortion parameters, the focal length, and projection type of the camera's lens model as extracted from the metadata [25]. Subsequently, the calibration parameters were used as input for processing the images of the whole study area.

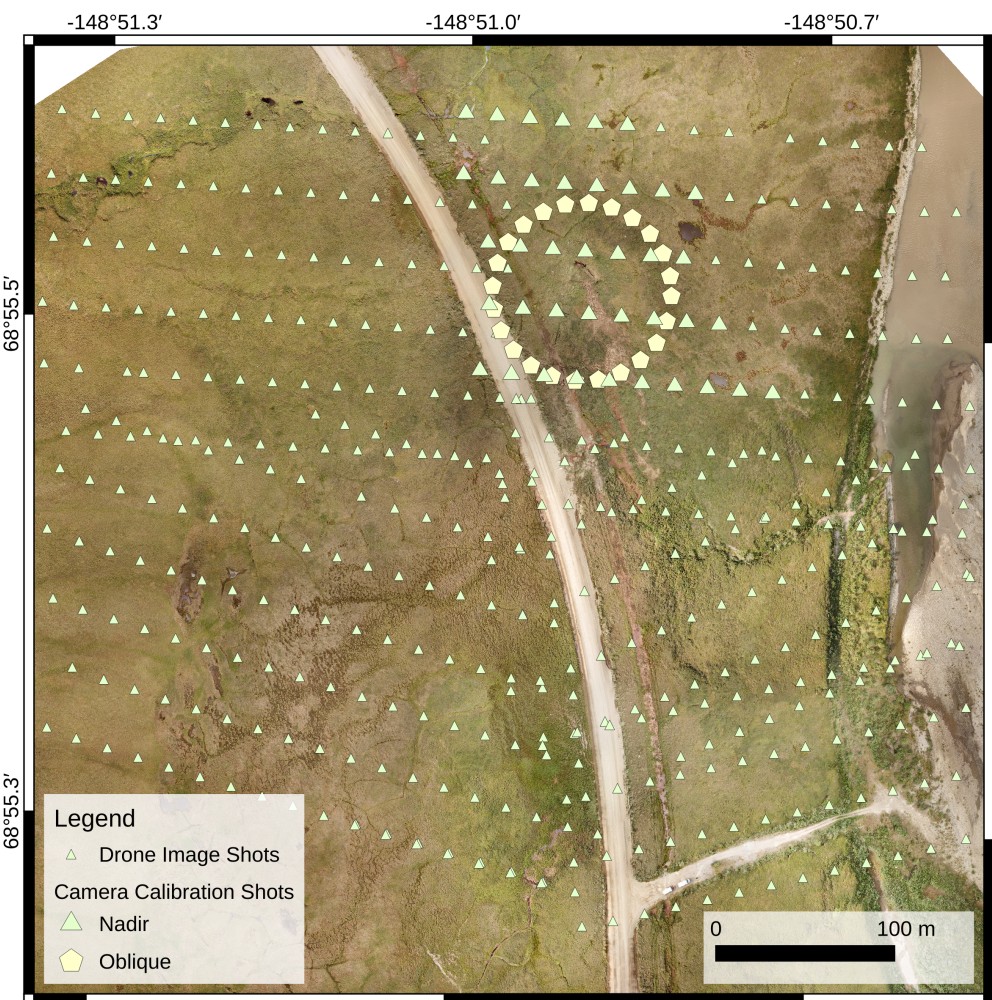

**Figure 3.** Positions of drone image shots in the study area. Acquisitions were collected at nadir and a height of 80 m a.g.l., with an overlap of 75% along-track and 65% across-track. The circular shape in the upper right corner depicts the drone camera positions during the point of interest flight at a height of 60 m a.g.l. and an image acquisition angle of 45°. As the point of interest we set the erosion feature. The bigger triangles and pentagons represent the images taken for camera calibration. Background image is the orthophoto mosaic (RGB) processed in OpenDroneMap.

### 2.3.2. Photogrammetric Processing

We produced georeferenced point clouds, orthophoto mosaics, and elevation models from the raw aerial images using the open cloud platform WebODM Lightning [28]. WebODM Lightning provides a user interface for the underlying photogrammetry software OpenDroneMap (ODM) [29]. OpenDroneMap is a free and open-source photogrammetry software designed for processing drone imagery and has a rich user and developer base. The processing pipeline of ODM is described in the official guide to the software [25] and can be seen in detail in Table A1.

For our study, we largely used the default settings for high-resolution photogrammetry products to retrieve the desired georeferenced point clouds. Table 2 gives an overview of the modifications made to the default settings based on the study site characteristics.

**Table 2.** Modified WebODM Lightning default settings used for the study area photogrammetry products development derived from field measurements, observations and the official OpenDroneMap software guide [25]. Copyright © 2019 by Piero Toffanin.

| Parameter | Explanation | Value |
|---|---|---|
| –dsm | Generates a Digital Surface Model (including vegetation). | True |
| –dtm | Generates a Digital Terrain Model. For distinguishing between ground and non-ground objects ODM uses a simple morphological filter (SMRF) [30]. We adjust the SMRF parameters in order to properly present the lowland tundra environment, see smrf-threshold, -window, -scalar and -slope [31]. | True |
| –pc-classify | The activation of pc-classify is mandatory to generate both, a Digital Terrain Model and Digital Surface Model. It enables parameter tweaking of SMRF to distinguish between ground- and non-ground objects [25]. | True |
| –smrf-threshold | Set to the minimum height (in m) of non-ground objects, which in our study area are mostly shrubs (Default: 0.5 m). | 0.2 |
| –smrf-window | Set to the size of the largest non-ground object (in m). Kept at the recommended minimum value of 10 m (Default: 18 m). | 10 |
| –smrf-scalar | Describes the dependence between the threshold and slope. It is recommended to increase this value slightly when minimizing the default smrf-threshold value (Default: 1.25 m). | 1.325 |
| –smrf-slope | Describes terrain slope of study area. Derived from the ratio between elevation change and horizontal distance change. The value 0.05 corresponds to a change of 50 cm over a 10 m distance (Default: 0.15). Differential GPS data of the study area show a maximum elevation change of 20 cm over a 10 m distance. To also account for the Dalton Highway we chose a value of 0.05. | 0.05 |
| –depthmap-resolution | Refers to an image containing information on the distance between camera and object: dark parts of the image are closer, bright parts farther away from the camera. Images are used to generate the point cloud. With an increasing value, the level of detail increases but also the occurrence of noise. We chose a value in between the default (640) and maximum (1000 [28]) to produce a high density point cloud while at the same time trying to mitigate a high amount of noise. | 800 |
| –use-opensfm-dense | Enables the use of the Python library OpenSfM for the Multi-View processing step (opposed to the default using the software suits Multi-View Environment). The activation of use-opensfm-dense is mandatory when defining the depthmap-resolution [25]. | True |

OpenDroneMap automatically calculated the ground sampling distance (GSD) from the flying height above ground, the cameras focal length and sensor width, and the image dimension, and processed the data accordingly. The results were georeferenced orthophotos, digital elevation models, and highly detailed 3D point clouds (see Table 1).

### 2.4. Point Cloud Processing

For all point cloud-related processing steps, we used the free and open-source 3D point cloud processing software CloudCompare [32]. The program contains all relevant analysis tools ranging from pre-processing (denoising, statistical outlier removal, etc.) to advanced point cloud registration algorithms and change detection tools [33].

#### 2.4.1. Denoising

Point clouds with detailed information are often subject to noise. These outliers need to be removed before a change detection analysis can be performed. CloudCompare offers a noise filter that—in contrast to the common statistical outlier removal filter—performs the analysis based on the distance of the points to the underlying surface instead of the distance to the neighboring points [34]. The low-pass filter implemented in CloudCompare estimates the underlying surface and subsequently removes points that are considered not to be part of the fitted plane [34]. The filter offers two options: filtering points by

a ball radius and filtering by a number of surrounding points. We tested both options. As a suitable value for the ball radius option was automatically detected by the program, we only needed to test for an appropriate number of nearest neighbors (kNN) around each point that was used to fit a plane into the cloud. Values ranging from 4 to 6 neighbors were tested on a transect of the study area, incorporating the infrastructural elements and the surrounding tundra. A kNN value of 4 proved to be suitable for the study area and resulted in a substantial decrease in point cloud density and storage volume.

### 2.4.2. Registration

Due to the lack of fixed reference points in the study area, an absolute image registration was not possible. We, therefore, aligned the point clouds using the Iterative Closest Point (ICP) Tool in CloudCompare. The ICP algorithm works under the assumption that both point clouds are already approximately aligned and that they depict the same features or are of similar shape [35]. Furthermore, we evaluated the accuracy of two image alignment approaches demanding different experience and intervention by the user.

User scenario 1 (supervised alignment)

The iterative closest point algorithm was applied on segments of the point cloud whose land surface is considered stable (with no visible changes). We, therefore, chose to remove the parking vehicles and the erosion feature which we detected during a field campaign in 2018 and 2019. The land surface area (visibly) unaffected by change was aligned with the ICP algorithm. The resulting transformation matrix was then applied to the whole cloud to compare it to the reference cloud. This procedure works best for experienced users capable of identifying land surface changes and with profound knowledge of the processing of 3D photogrammetry products.

User scenario 2 (unsupervised alignment)

The ICP algorithm was applied on the whole point cloud. This procedure may be suitable for users with little to no training in the processing of 3D photogrammetry products or if clear sings of land surface changes are not visible. The fine-tuning input parameters of the ICP alignment were set as follows:

- The iteration process was set to stop when a root mean square error (RMS) difference of $10^{-5}$ m was reached. This small threshold value comes with a high computation time but a very accurate alignment [35].
- In 2019, the UAV captured slightly more images. We therefore set the theoretical overlap of the clouds for user scenario 2 at 85% (90% of the area extent in 2019 were covered in 2018, an additional 5% were set as a an uncertainty buffer). Due to the exclusion of the erosion feature and the vehicles the overlap was set at 80% for user scenario 1.
- CloudCompare aims at optimizing the computation speed. Therefore, it implemented a random sampling limit that describes a maximum number of points which are randomly sub-sampled from the cloud for each iteration [35]. The limit was set to 70,000 points with the farthest point removal enabled. The latter led to an elimination of points that are too far away from the reference cloud.

In addition, we aligned the 2019 point cloud to the 2018 point cloud and vice versa and created two subsets of the area for a thorough performance assessment. One subset contained the erosion feature and another contained a stable area of the Dalton Highway (see Figure 1B,C).

### 2.5. Repetition with AROSICS

After the first processing of the data following the proposed workflow, a visual inspection of the processed point clouds showed that the ICP was not as successful in correcting the horizontal offset as it was in improving the vertical one. We, therefore, decided to insert a further processing step before the ICP alignment. Using the global image co-registration of Python library AROSICS [36], we derived a x-/y-map shift on the

basis of the orthophoto mosaics, applied it on the point cloud, and started over with the step of the ICP.

### 2.6. Ground Truthing

Our proposed workflow aims at deriving sub-meter land surface changes by using consumer-grade UAVs without applying RTK measurements or GCP. However, to validate our results, we distributed checkerboard GCP targets with a dimension of 50 by 50 cm evenly in the study area and measured their center points with a differential GPS. As permanent GCP are missing in our study area, we selected an access road leading from the Dalton Highway to the Sagavanirktok River to set up the dGPS base. Unfortunately, due to a firmware update of the dGPS after the first acquisition in 2018 the profile, we established for the measurements could not be reproduced in 2019. The median elevation of both acquisition years depicts an offset of 1.69 m. A horizontal offset was impossible to derive because neither the checkerboard GCP targets nor the base station were set up at the exact same spots. Therefore, we used the acquired dGPS data for a relative ground truthing: we chose the first checkerboard GCP target measurement of 2018 as a reference point and calculated the distance and elevation difference to every other checkerboard GCP target in 2018. We repeated this for 2019. This allows us to compare the dGPS data with the point cloud output from OpenDroneMap and validate if they show the same relative differences in their horizontal and vertical positions.

### 2.7. Change Detection

We based our change detection analysis on the comparison of the 2018 and 2019 point clouds. In contrast to analyzing land surface changes based on gridded DEM of difference (DoD) [37], point clouds allow a more detailed representation of complex landscapes; point clouds capture vertical and overhanging elements without being restricted to the spatial resolution of the DEM or without elements being omitted during a gridding process [38]. Capturing these elements at a high-resolution is crucial when looking, for example, at gully formation and polygonal patterns in the Arctic. In our study, we applied the Multiscale Model to Model Cloud Comparison (M3C2) algorithm by Lague et al. [38], which is implemented in CloudCompare. The M3C2 approach bases its calculations on a set of core points, which are a sub-sample of the reference cloud. Within a given radius around each core point, a surface normal is calculated by fitting a plane to all its neighboring points [39]. The radius is set in dependence on the local roughness and density of the reference cloud [38–40]. In CloudCompare we used the "Guess params" option in order to estimate a suitable radius value. Once the normal was defined, the algorithm computed the distance between the reference and the compared cloud along the latter. A positive distance value means the compared cloud is above the reference cloud, a negative value states it is below [40]. This translates to land surface uplift (e.g., due to sedimentation) indicated by positive values, whereas land surface subsidence (e.g., due to erosion) is indicated by negative values. Every core point was assigned the information on the distance (in meters), its uncertainty (a value ranging from 0 to 1), and the normal vector (Nx, Ny, Nz).

In contrast to other cloud comparison algorithms, M3C2 shows lower error estimates and good tolerance in clouds with a high surface roughness [39]. It can also be applied to images containing non-overlapping areas such as those in our case [41], where the acquired area in 2019 was slightly larger. A detailed description of the M3C2 algorithm is provided by Lague et al. [38].

### 2.8. Change Projection

To further evaluate the impact that the detected land surface changes may have on nearby infrastructure already or in the near future, we evaluated the spatial direction of the land surface displacement. By multiplying the normal vector and the M3C2 distance, we derived the total displacement vector and further calculated the magnitude of the displacement in the vertical and horizontal directions. The determined displacement vector in 3D (Dx, Dy, Dz) and 2D (Dx2D, Dy2D) and their magnitude were stored as an additional

attribute for every point. These and all further calculations were conducted in Python (version 3.6). During this procedure, all changes in the land surface were calculated for the individual core points at a very high spatial resolution. In order to analyze where coherent regional displacements occurred in the study area, a reduction and spatial aggregation of the high-resolution point data was necessary. Therefore, we generated a raster of the vertical displacement (Dz) of the point cloud by using inverse distance weighting (IDW). For the raster output, we set spatial resolutions of 2.5, 5, 10, and 20 m per pixel to identify a suitable spatial resolution that eliminates low-level noise and artifacts while at the same time capturing relevant spatially coherent land surface changes. Subsequently, we applied the Sobel edge detection algorithm on the raster image to retrieve locations of high image gradients. A high gradient symbolizes neighboring pixels with high differences in values (from a very low to a very high vertical displacement and vice versa). The raster image of the gradient was the basis for the subsequent application of the kMeans algorithm that clusters data into categories depending on their similarities [42]. We chose two classes: change (a particularly high gradient) and no change (a particularly low gradient). We retrieved a binary raster image outlining areas of change and no change and constructed a vector overlay to mask the corresponding points from the complete cloud.

## 3. Results

### 3.1. Accuracy Assessment

In this section, we assess the performance of the tested denoising algorithms and the point cloud registrations performed with ICP and AROSICS, for the whole study area and the defined point cloud subsets.

### 3.1.1. Whole Study Area

The point clouds from 2018 and 2019 as generated by ODM showed an average offset of 1.2 m horizontally (measured at the culverts in the center of the study area) and a substantial vertical offset of 7.0 m (derived for the whole study area). The denoising with the kNN4 algorithm reduced the size of the point cloud from over 70 and 84 million points by 27–30% for 2018 and 2019 (see Table 3), respectively. The subsequent ICP alignment resulted in an RMSE of 0.98–1.07 m (see Table 4).

**Table 3.** Number of points in raw point cloud (post-processing level I) and after denoising (post-processing level II) in CloudCompare. We tested two low-pass filter options: (i) filtering points by a ball radius, which is automatically detected by the software and (ii) filtering by a number of 4 neighbouring points (kNN4).

|  | 2018 |  | 2019 |  |
| --- | --- | --- | --- | --- |
| **Point Cloud** | **Number of Points** | **(%)** | **Number of Points** | **(%)** |
| Raw | 70,574,548 | 100 | 84,731,227 | 100 |
| Denoised |  |  |  |  |
| — ball radius | 60,746,799 | 86 | 68,806,542 | 81 |
| — kNN4 | 49,260,479 | 70 | 61,522,497 | 73 |

**Table 4.** Root Mean Square Error [m] of the alignment under post-processing level III (ICP alignment only) and post-processing level VI (AROSICS and ICP alignment).

| **Post-Processing Level III** | **Reference Cloud 2018** | | **Reference Cloud 2019** | |
| --- | --- | --- | --- | --- |
|  | **kNN4** | **Ball Radius** | **kNN4** | **Ball Radius** |
| unsupervised alignment | 1.07 | 1.12 | 1.05 | 1.07 |
| supervised alignment | 1.00 | 1.05 | 0.98 | 1.00 |
| **Post-Processing Level IV** |  |  |  |  |
| unsupervised alignment | – | – | 1.05 | – |
| supervised alignment | – | – | 0.99 | – |

Applying the ball radius, on the other hand, the point cloud remained at a high density (81–86% of the original number of points, see Table 3) and the point cloud alignment was performed with a higher error (RMSE of 1–1.12 m, see Table 4). Focusing on the results of the kNN4 denoising algorithm, the lowest RMSE of 0.98 m was achieved with the combination of a supervised alignment (excluding the erosion feature and vehicles) and referencing the 2018 to the (denser) 2019 cloud (see Table 4). The ICP alignment successfully minimized the vertical offset between the two point clouds; from an original average elevation difference of 7.0 m in the whole study area, the alignment reduced this vertical offset to 0.7 m. During the development of the workflow, however, a visual inspection of the point clouds showed that the ICP alignment was deficient in correcting the horizontal offset (see Figure 4C). Using the image registration of AROSICS on the orthophoto mosaics, we detected a shift of −0.8 m and +0.9 m in a x- and y-direction, respectively. Applying the calculated x-/y-shift on the point cloud and repeating the ICP alignment, finally, minimized this horizontal offset to an average of 0.2 m (see Figure 4D) with a RMSE of 0.98 m (Table 4).

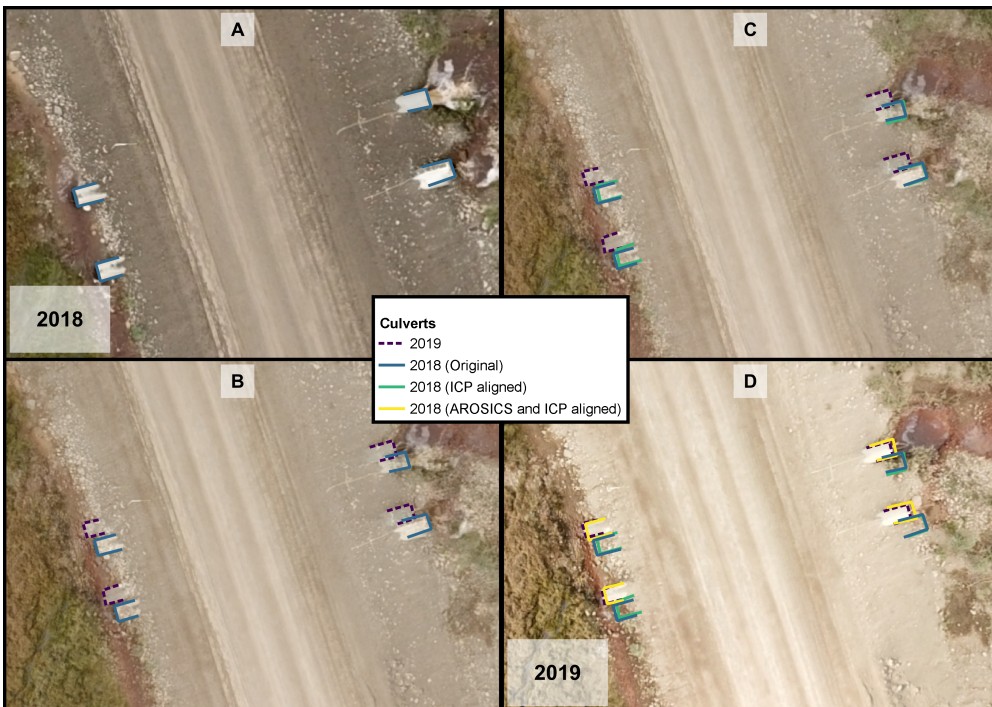

**Figure 4.** Results of the co-registration digitized along the culverts which drain the area west of Dalton Highway: (**A**) position of the culverts digitized on the basis of the original 2018 orthophoto mosaic (blue) and; (**B**) additional position in 2019 as reference (purple dashed line); (**C**) 2018 after ICP correction (green) and; (**D**) 2018 after AROSICS and ICP correction (yellow). (Background images: (**A**) orthophoto mosaic 2018, (**B**,**C**)) orthophoto mosaic 2018 and 2019 overlaid with 50% transparency each and (**D**) orthophoto mosaic 2019).

In order to also evaluate the quality enhancement of the point cloud comparison achieved by the denoising and combination of registration algorithms, we present the results of the M3C2 distance after every processing step of the workflow: post-processing level I represents M3C2 distances calculated by comparing the raw point clouds (as computed by ODM), level II is calculated based on the denoised point clouds, level III entails the denoising and the ICP alignment (as proposed in the initial workflow design), and level IV represents the values after incorporating the additional horizontal offset correction with AROSICS along with denoising and the ICP alignment.

Under post-processing levels I and II, we derived a median M3C2 distance of 6.60 m and 6.61 m when comparing the point clouds from 2018 and 2019. This would suggest a major land surface uplift of the entire study area. After post-processing level IV, which applied the point cloud alignment with ICP and AROSICS, however, the calculated M3C2

distance decreased to 0.02 m indicating that consistent terrain topographies were achieved for 2018 and 2019 for the study area.

### 3.1.2. Subsets of Study Area

A more detailed picture of the performance of the workflow was retrieved by examining the point cloud subsets. In the point cloud subset of the stable area around the junction of the Dalton Highway (see Figure 1C), we were able to substantially reduce the number of outliers in comparison to the raw data set by applying the denoising with kNN4: of the original 21,929 points of the cloud subset, 14,744 points (33% less) remain (see Figure 5A level II of post-processing). Evaluating the quality of the post-processing steps in the image subset of the erosion feature confirmed this result. A considerable reduction in the noise with a simultaneous reduction in the number of points by 34% (from 32,720 points to 21,571 points) can be seen (Figure 6A Level II).

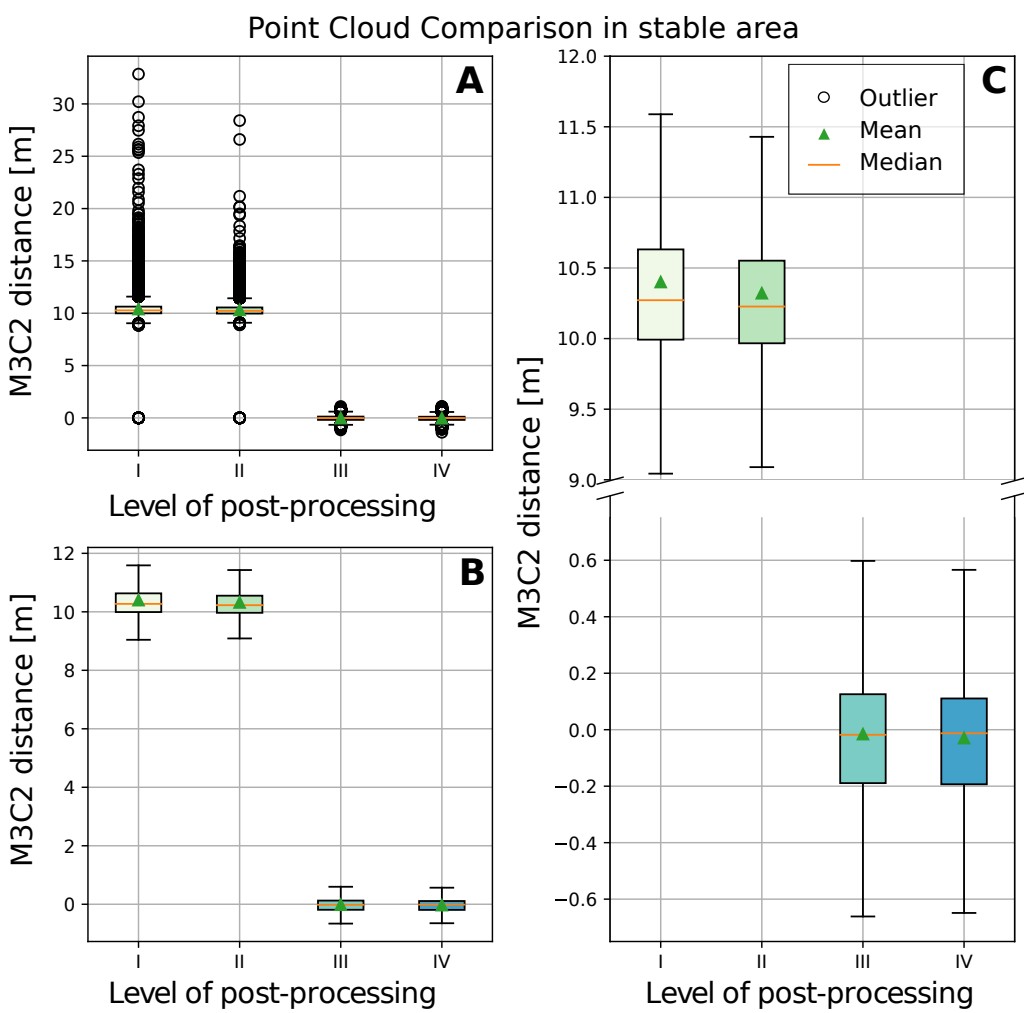

**Figure 5.** Cloud distances derived via Multiscale Model to Model Cloud Comparison (M3C2) in the point cloud subset of the stable area at the junction of the Dalton Highway. Results are grouped according to the levels of post-processing (I) raw, (II) denoised (III) denoised + ICP aligned, (IV) denoised + AROSICS shifted + ICP aligned data. Boxplots (25% and 75% percentile) (**A**) with and (**B**) without outliers and (**C**) close-up on median, mean and percentile values.

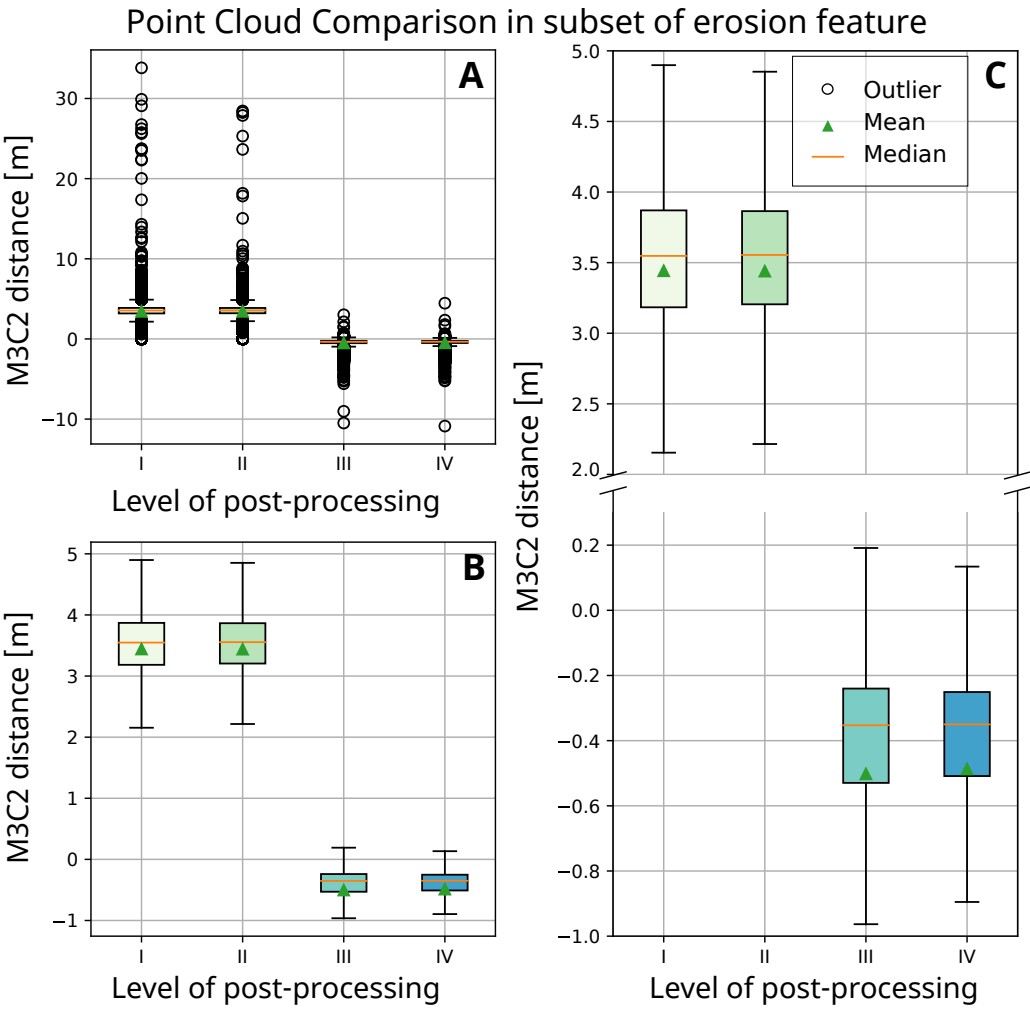

**Figure 6.** Cloud distances derived via Multiscale Model to Model Cloud Comparison (M3C2) in the point cloud subset of the erosion feature. Results are grouped according to the levels of post-processing (I) raw, (II) denoised (III) denoised + ICP aligned, (IV) denoised + AROSICS shifted + ICP aligned data. Boxplots (25% and 75% percentile) (**A**) with and (**B**) without outliers and (**C**) close-up on median, mean and percentile values.

The subsequent alignment with ICP corrected the vertical offset between the 2018 and 2019 point clouds. Under post-processing levels I and II we derived a median M3C2 distance of around +10.2 m, which again would indicate a major land surface area uplift in the subset of the stable area. After the ICP alignment the median value was −0.02 m, translating to no land surface change as already derived in the whole study area (see Section 3.1.1 and Figure 5C). The IQR had also decreased significantly. The values decreased from 0.64 m and 0.58 m under post-processing levels I and II, respectively, to 0.32 m in level III. The same applied to the results in the subset of the erosion feature. The ICP alignment corrected the vertical offset and improved the results of the M3C2 distance values: the median decreased from originally +3.5 m (land surface uplift/ sedimentation) to −0.35 m (land surface subsidence), the IQR was reduced to 0.29 m (from 0.68 m under the level I), see Figure 6B.

The most pronounced improvement was achieved under post-processing level IV. As mentioned above, implementing the additional AROSICS registration reduced the horizontal offset to 0.2 m. The median M3C2 distance in the subset of the stable area was further reduced to −0.01 m after the joint AROSICS and ICP correction, and the IQR decreased to 0.30 m (see Figure 5C level IV). In the subset of the erosion feature, the median M3C2 distance value remained at −0.35 m.

### 3.1.3. Ground Truthing

As described in Section 3.1, post-processing level IV achieved the highest accuracy when comparing the point clouds (see Table 4 and Section 3.1.1). The pre-processing included denoising both point clouds with kNN4, correcting the horizontal offset with AROSICS, and finally, ICP aligning the 2018 to the 2019 cloud under user supervision (see user scenario I in Section 2.4.2). Comparing the 2018 point cloud at post-processing level IV with the 2018 dGPS data, we confirmed this result. The raw ODM Output (post-processing level I) showed an RMSE of 0.68 m and 3.3 m of the relative horizontal distance and vertical difference, respectively. The values deviated, especially along the edges of the study area, with an RMSE of 0.91 and 4.31 m for the horizontal distance and vertical difference (over 0.39 and 2.07 m in the center). We minimized both values further to a RMSE of 0.58 m and 3.12 m after post-processing level IV for the whole study area (see Figure A1A,B). The improvement was also considerable along the image edges. The RMSE of the relative distances and differences decreased from 0.91 and 4.31 m to 0.8 and 3.95 m, respectively. As we were aligning the 2018 to the 2019 point cloud, we defined the dGPS data of the 2019 acquisition as the reference for a final quality assessment. We compared the 2019 dGPS data with the 2019 ODM point cloud output and derived a transformation matrix that provides information about the point cloud rotation and translation required to achieve consistent coordinate system origins. This transformation was applied not only to the 2019 point cloud but also to the 2018 point cloud after post-processing level IV. We subsequently repeated the comparison of the 2018 to the 2019 point cloud (now translated and rotated with the above-mentioned transformation matrix) to validate our relative measurements. The absolute ground truthing revealed that we had achieved an RMSE of 0.55 m and 0.65 m for the relative horizontal distances and vertical differences, respectively (see Figure A1C,D). The highest errors occurred along the image edges, with RMSE values of 0.75 and 0.86 m versus 0.27 and 0.39 m in the center of the image (for the horizontal distances and vertical differences, respectively). Nonetheless, the median M3C2 distance value remained at 0.02 m for the whole study area and −0.35 m in the subset of the erosion area. Only the value in the subset of the stable area changed slightly from −0.01 m to −0.03 m.

### 3.2. Change Detection and Projection

After rasterizing the point cloud to spatial resolutions of 2.5, 5, 10, and 20 m, we identified a pixel size of 5 × 5 m as most suitable for our study. In combination with the edge detection filter and subsequent clustering, we could successfully eliminate noise and artifacts of the point cloud construction and identify those areas showing consistently high vertical displacement (see Figure 7A). We observed most of the changes along the riverbank and riverbed. By comparing the 2018 and 2019 imagery, we detected changes in water level. In 2018, the riverbed was largely covered by water from the Sagavanirktok River. In the 2019 imagery, it was exposed to a great extent (as seen on the right edge of the study area in Figure 7A). Furthermore, our workflow detected a vertical displacement along a gully (see Figure 7A,2). A visual examination of the images confirmed the auto-detection. Here, we observed signs of an incision due to the drainage into the river; the formerly vegetated area was exposing sediments in the second year. The workflow further detected a vertical displacement along another gully structure in the upper left corner of the study area (Figure 7A,3). We were not able to verify this via the visual comparison; however, we were able to confirm that the gully carried more water in 2018 than in 2019, likely leading to some shallow parts being connected and slightly inundated, which might have led to the high image gradient. Moreover, we observed a shift to the north of the gully structure due to a consistent image offset, which was greatest at the edges of the image. Throughout the whole study area, we further detected several isolated high gradients in vertical displacements with a dimension of one (5 × 5 m), two (5 × 10 m), or four by one (5 × 20 m) pixels. A visual examination revealed that two of these supposed displacements originated from ponding water during the 2018 acquisition that had vanished during the

2019 acquisition. The remaining displacements appeared along the edges of the image, where we already observed a consistent offset. The isolated high gradients at the northern edge result from an offset to the northeast of polygon structures and the trench-like structure parallel to the highway. At the southern edge, we noticed an offset of individual bushes and shrubs to the southwest together with the embankment of the Dalton Highway, again pointing to the presence of distortion errors at the edges of the image. Visually, however, we cannot confirm true land surface changes in these areas. Thus, we consider them artifacts of the registration. The erosion feature (Figure 7A,1) was successfully detected over the whole study area. As described in Section 3.1.2, the calculated median M3C2 distance for this specific landscape element was −0.35 m.

Choosing a lower spatial resolution of 10 m showed a similar filtering result with fewer artifacts originating from the image offset along the southern edge (see Figure A2C). However, it lacks spatial detail, which is needed for projecting the development of the erosion feature. Resampling to 20 by 20 m per pixel led to an exclusion of all relevant land surface displacements except for the changes along the riverbank (Figure A2D). Selecting a spatial resolution of 2.5 m, we observed a substantial increase in artifacts from the image offset and in the number of areas that were inundated during the 2018 acquisition but dried up in 2019, see Figure A2A.

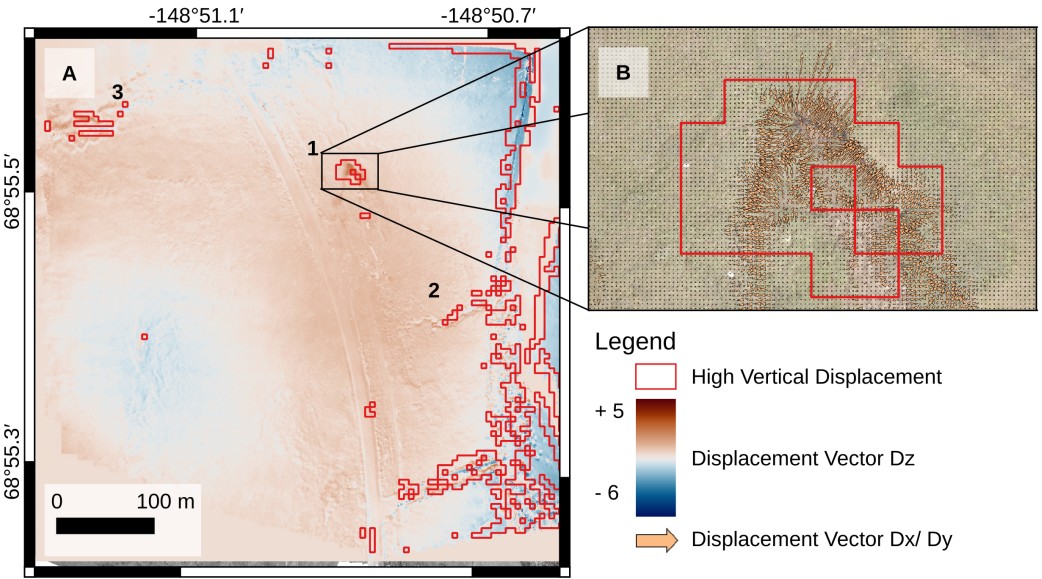

**Figure 7.** (**A**) Vertical displacement vector Dz in the study area. The overlying pixel mask shows areas with a high image gradient, translating to areas with a high difference in vertical displacement values. (**B**) presents the derived directions of the erosion feature and their magnitude as represented by the length of the arrows. Background image in subset (**B**) is the orthophoto mosaic of 2019 as processed in OpenDroneMap.

To project the future development of the erosion feature, we applied our pixel mask on the processed point cloud to extract the areas of a high vertical displacement. We focused on the extent of the known erosion feature, which was detected with a dimension of 20 pixels (see Figure 7A,1), a maximum width and height of 25 and 20 m, respectively. The Sobel algorithm identified three pixels west of the erosion feature (see Figure 7B) with a supposedly high vertical displacement, which visibly do not show signs of land surface change. Examining the raster of the vertical displacement (Dz) confirms the clearly smaller dimension of the erosion feature with a maximum width and height of only 15 by 20 m. The Sobel edge detection algorithm emphasizes the gradients in an image, leading to an incorporation of adjacent pixels simply because they show much higher or lower values than their neighboring ones. This circumstance is very useful as it allows us to distinguish between artifacts (of 1–2 pixels) and actual land surface changes. In intervals of

0.5 m, we analyzed the displacement vector Dx and Dy providing us with the magnitude and direction of the movement of each point and found a diverging movement of the erosion feature (see Figure 7B). While the divergence is relatively homogeneous at the center, we observe a strong expansion to the NNE at the head wall of the erosion feature. These directions run along an area that shows clear polygonal structures in the orthophoto mosaics, indicating active degradation of the ice wedges. During the 2018 acquisition, we noticed a high amount of standing water covering the area from the culverts to the erosion feature.

## 4. Discussion

The proposed workflow substantially improved the quality of the point cloud comparison. By denoising the high-resolution point clouds with kNN4, we eliminated outliers and reduced the number of points to 70% of the original amount. Despite the self-calibration of the lens post-flight (see Section 2.3.1), we observed an overestimation of a depression west of the Dalton Highway in the 2018 processed point cloud. This, in combination with the less accurate on-board GPS sensor, led to the high elevation offset of 7.0 m as described in Section 3.1.1. However, using the ICP substantially reduced this relict of parallel flight path acquisition to a minimum of 0.7 m vertically. Adding the step of the AROSICS registration beforehand further decreased the horizontal offset by a factor of six (from 1.2 m under post-processing level III to 0.2 m under level IV). Reducing the horizontal offset is essential when evaluating directional land surface changes that potentially pose a threat to infrastructure such as the Dalton Highway. The visual examination revealed that the AROSICS correction was very efficient but mostly along the culverts in the center of the image. AROSICS successfully identified these structures as valuable tie points. Reaching the edges of the image, the recognition of suitable tie points became more challenging due to the lack of unique features or sharp contrasts of tundra landscapes. This led to a consistent offset of several meters along the edges of the image.

This explains why even after both corrections we encountered an RMSE of 0.99 m for the 3D alignment (see Table 4). This RMSE result is lower than accuracies derived in other UAV-based studies on land surface changes, where the error of the photogrammetric product usually ranges in the scale of a few centimeters [14,43]. These studies, however, rely on images acquired with RTK technology, the use of GCP, or complimentary airborne or terrestrial laser scanning data, which mostly are not available to local environmental observers or citizen scientists conducting ad-hoc surveys with low-cost consumer-grade UAVs. Our workflow, therefore, opens a pathway for using such lower-accuracy UAV survey techniques in science and engineering applications focusing on terrain change detection.

In its current state, the proposed workflow also successfully detected and quantified an erosion feature in close proximity to the Dalton Highway. The raster of the vertical displacement and the subsequent clustering with kMeans depicted spatially coherent land surface changes. The erosion feature and the progression of a gully draining into the Sagavanirktok River (see Figure 7A,1,2) were detected. We also gained insight into water level changes in the river and individual pooling of water throughout the study area. The supposedly eroding gully in the northwest corner of the study area, however, was an artifact of the consistent offset at the image edges. A shortcoming that, again, can be minimized by the use of different flight patterns. The detected land surface displacement of $-0.35$ m is slightly higher as reported in other studies conducted in Alaska, where annual subsidence values usually stay below 0.10 m [44–46]. With a root mean square error still at 0.99 m and a displacement value of $-0.35$ m, we can conclude that in its current state, the workflow is not suitable for detecting seasonal or slow and gradual land surface changes. Assuming the displacement rate persists, it will be able to retrieve an erosion signal exceeding the registration uncertainty only after three years. However, it provides a valuable detection and quantification of rapid and abrupt land surface displacements as well as water level changes.

Assessing the direction and magnitude of the erosion feature, we observed an expansion in an NNE direction. This trajectory was consistent with recognizable patterned

ground structures in the area, and the observations indicate that the degradation followed the ice wedge polygonal network. As we only investigated one time step, the derived direction and magnitude represent only a very narrow snapshot summarizing one year of change, while the development of the erosion feature may deviate considerably in the future. Assuming the erosion process continues along the polygonal rims of the ice wedge network, it is likely that there will be a change in direction: the rims in close proximity to the feature diverge to the north and southwest (towards the highway). During the 2018 image acquisition, we further observed a great amount of water pooling at the erosion feature which can accelerate the permafrost degradation [7,47,48]. The future development is based on the simplified assumption that the observed displacement continues linearly. An accurate projection would, however, need observations over longer time periods and complementary data on precipitation, hydrology, morphology, and ground ice distribution in the study area, as they have a substantial impact on the rate and growth direction of permafrost degradation processes.

Our workflow has the clear advantage that it does not rely on a cost and time-intensive setup with dGPS measurements. Designing the workflow with free and open-source software allows anyone to perform monitoring of land surface changes at no additional costs and on any available operating system. Another benefit is the use of off-the-shelf UAVs. They are easy for anyone to operate and therefore provide an affordable opportunity for local citizens and other stakeholders interested in monitoring and documenting changes in the tundra environment related to global warming or other disturbances. Off-the-shelf UAVs could encourage more research and data collection conducted by local environmental observers using a simple UAV setup. Designing the different user scenarios, we found that the unsupervised alignment achieved a comparable alignment accuracy with an RMSE of 1.05 m. This underlines the potential of the workflow to be used by, e.g., citizen scientists [49]. Still, an improvement in the accuracy of the point cloud registration is needed to retrieve seasonal and slowly progressing land surface displacements. To enhance the self-calibration of the camera lens, we therefore propose to acquire complementary oblique images with a perpendicular flight path. This will prevent doming effects [26] or the overestimation of depressed landforms. Conducting these additional surveys will increase the time needed during field campaigns but delivers a higher alignment accuracy. For improving point cloud comparisons, we recommend implementing the correspondence-driven plane-based M3C2 approach by Zahs et al. [50]. Their method provides a seven-fold reduction in the uncertainty when computing sub-meter changes of land surfaces that have a high roughness.

## 5. Conclusions

Our study showcases the successful use of off-the-shelf, consumer-grade UAVs without real-time kinematic GPS correction and without fixed ground control points to detect and quantify short-term and sub-meter ground surface changes attributed to permafrost thaw. We accomplished a workflow that highly improves the quality of the point cloud comparison via M3C2 by minimizing the point cloud noise and vertical and horizontal offset between the acquisition years. The workflow is entirely based on open-source and free software and a minimal drone setup, that is time and cost-efficient. With this, we successfully derived a land surface displacement at a site along the Dalton Highway with a median value of −0.35 m between 2018 and 2019 and determined its magnitude and direction. Due to the relatively high RMSE of 0.99 m during the point cloud alignment, we can recommend the use of the workflow in its current state for abrupt and rapid land surface displacements such as thaw slumping, gully erosion, and ice-wedge degradation. To use it for monitoring slowly progressing and seasonal land surface changes, we propose an enhancement by using alternative flight patterns and a modified M3C2 cloud comparison approach. We conclude that there always will remain a trade-off between fast, consumer-grade, and affordable UAV setups on the one hand and science/survey-grade highly accurate but cost- and labor-intensive measurements with RTK-corrected UAV, complimentary laser scanning, or the use of GCP on the other hand. Despite the

current limitations, our workflow shows potential for change detection applications using drone imagery, which was acquired without oblique images at perpendicular flight paths and complimentary RTK measurements. The detection and quantification of land surface changes with similar rates will be most accurate if the acquisitions are on a multi-year basis and once it exceeds the registration uncertainty of three years.

**Author Contributions:** Conceptualization, S.K., G.G., J.B. and M.L.; methodology, S.K., G.G., J.B. and M.L.; software, S.K.; validation, S.K.; formal analysis, S.K., G.G., J.B. and M.L.; resources, M.L.; data curation, S.K. and M.L.; writing—original draft preparation, S.K.; writing—review and editing, S.K., G.G., J.B. and M.L.; visualization, S.K.; supervision, G.G., J.B. and M.L.; project administration, M.L.; funding acquisition, M.L. and S.K. All authors have read and agreed to the published version of the manuscript.

**Funding:** This work was conducted within the young investigator group PermaRisk, which is funded by the German Federal Ministry of Education and Research (BMBF) under the funding reference number 01LN1709A. S.K. was additionally supported by a 2-year grant from Christiane Nüsslein-Volhard foundation and a stipend from family fund of Humboldt-Universität, Berlin.

**Data Availability Statement:** Data input, output, interim products and Python code are available from https://doi.org/10.5281/zenodo.7376621 (accessed on 28 November 2022).

**Acknowledgments:** We further acknowledge the support of the citizen science project UndercoverEisAgenten, which is funded by the German Federal Ministry of Education and Research (BMBF) under the funding reference number 01BF2115A.

**Conflicts of Interest:** The authors declare no conflict of interest. The funders had no role in the design of the study; in the collection, analyses, or interpretation of data; in the writing of the manuscript, or in the decision to publish the results.

## Abbreviations

The following abbreviations are used in this manuscript:

| | |
|---|---|
| AROSICS | Automated and Robust Open-Source Image Co-Registration Software |
| ICP | Iterative Closest Point |
| IDW | Inverse Distance Weighting |
| IQR | Interquartile Range |
| kNN | k-Nearest Neighbour |
| M3C2 | Multiscale Model to Model Cloud Comparison |
| ODM | OpenDroneMap |
| POI | Point of Interest |
| SfM | Structure from Motion |
| UAV | Unoccupied Aerial Vehicle |

## Appendix A

**Table A1.** Processing Pipeline of OpenDroneMap, modified after the official OpenDroneMap software guide [25]. Copyright © 2019 by Piero Toffanin.

| Processing Step | Explanation |
|---|---|
| Load Dataset: | ODM reads metadata from the EXIF (Exchangeable Image File Format) tags of all images, containing information on geolocation (GPS) |
| Structure from Motion: | The photogrammetry technique computes the camera's position and angle (camera pose) for every image by looking for unique features that are visible in both (or more) images [25]. This allows the creation of a sparse point cloud [25]. OpenDroneMap uses the Python library OpenSfM [51,52]. |

**Table A1.** *Cont.*

| Processing Step | Explanation |
| --- | --- |
| Multi-View Stereo (MVS): | Complementing the SfM technique, MVS uses the derived camera information and sparse point cloud to produce a highly detailed (dense) point cloud [25]. |
| Meshing: | In this step, the 3D points of the cloud are connected to form triangles forming a polygonal mesh. |
| Texturing: | The mesh, the camera poses and the images build the basis for the texturing. Every polygon of the mesh is assigned the best fitting image from which the colour is derived [25]. MvsTexturing is the software used by OpenDroneMap [53]. |
| Georeferencing: | This step contains the transformation of local coordinates to the actual geographic coordinates. The information on the world coordinate system is extracted from the images' GPS tags. |
| Digital Elevation Model Processing: | ODM uses the georeferenced point cloud and applies an inverse distance weighting interpolation (IDW) method to extract a surface model [25]. Missing values (gaps) in the model are filled within IDW interpolation and noise is filtered using a median filter [25]. |
| Orthophoto Processing: | In the last step the textured 3D mesh is loaded into an orthographic scene and is virtually captured and saved from above as an image. The result is a georeferenced orthophoto, cropped to its geolocation boundaries [25]. |

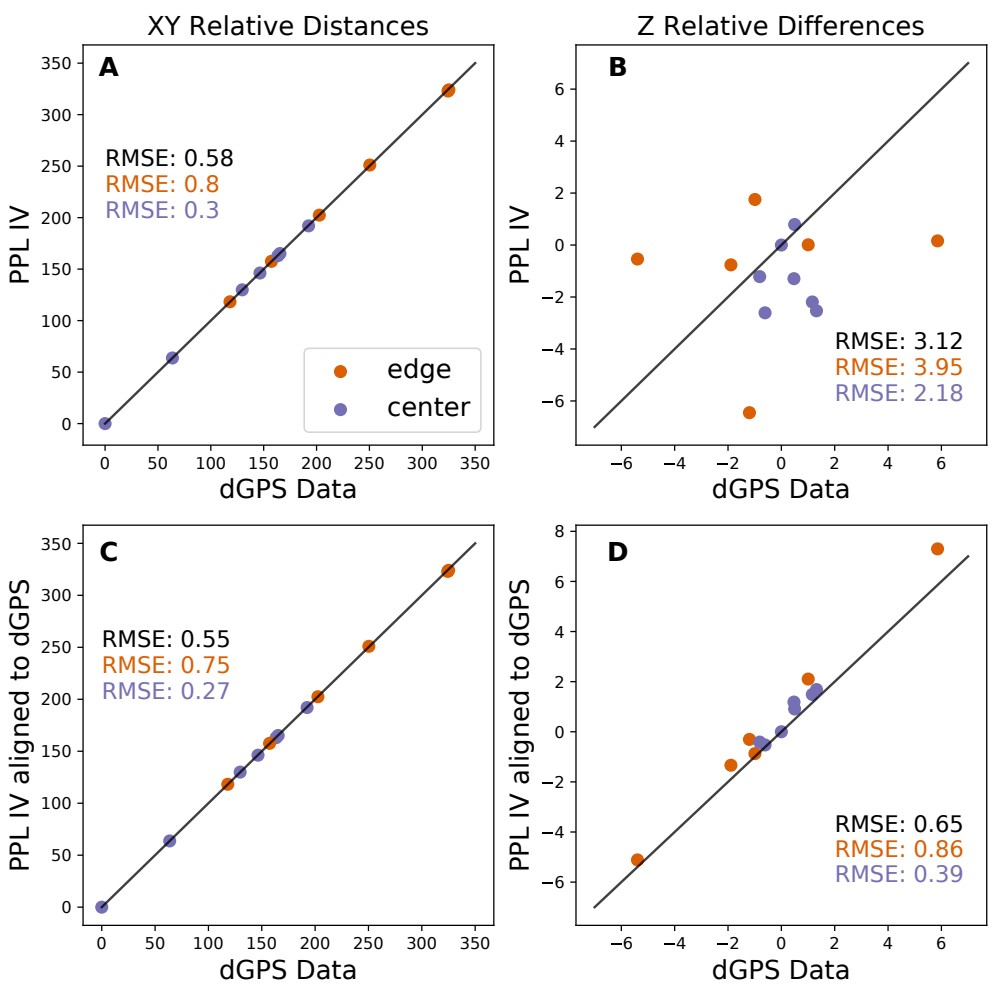

**Figure A1.** Comparison of relative horizontal distances and vertical differences with the ones of the point cloud at post-processing level IV (**A**,**B**) and after rotating and translating the point cloud at post-processing level IV with the 2019 transformation matrix (**C**,**D**).

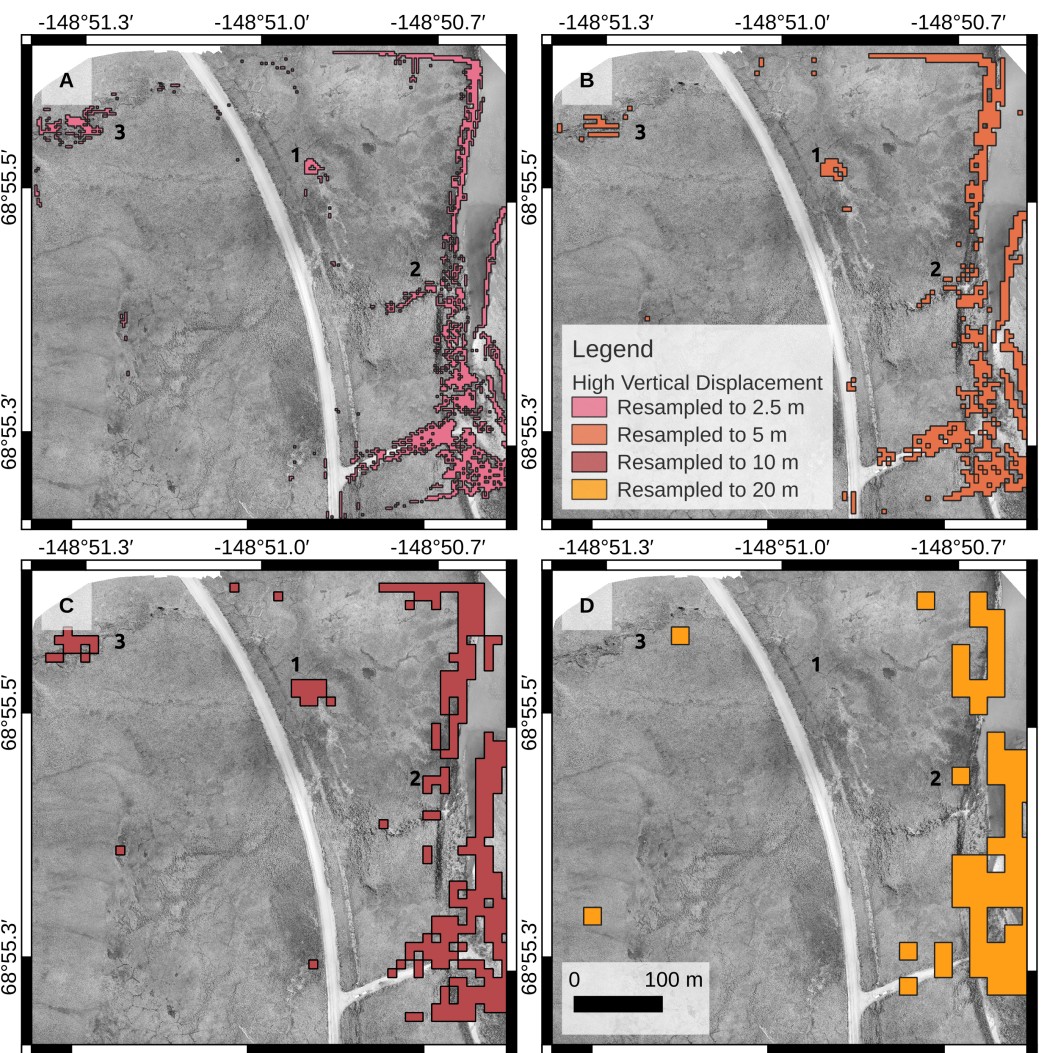

**Figure A2.** Areas with a high or low difference in vertical displacement values, after resampling to (**A**) 2.5, (**B**) 5, (**C**) 10 and (**D**) 20 m and subsequently applying the Sobel edge algorithm and kMean clustering.

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
