# Peer review of "The Potential of UAV Imagery for the Detection of Rapid Permafrost Degradation: Assessing the Impacts on Critical Arctic Infrastructure"

_remotesensing, doi:10.3390/rs14236107_

Round 1

Reviewer 1 Report (Previous Reviewer 1)

I have read the new corrected version of the article and I consider that it has improved compared to the previous version. The authors have considered all suggestions made by me. Therefore, I consider that the article is fit for publication.

Author Response

Reviewer 2 Report (Previous Reviewer 3)

I read through the revised manuscript and found the authors have responded to all of my concerns. The manuscript has been improved after the revisions. Therefore I recommend to accept it for publication at its present state.

Author Response

Reviewer 3 Report (Previous Reviewer 4)

General Comments

The authors have made improvements to the manuscript. It is interesting to recognize that each step of image processing affects the data. So, the minimal use of image processing techniques such as resampling, smoothing data, or using Sobel filters is important for retaining data quality and integrity. However, there are numerous grammatical errors that need improvement. These errors detract from an otherwise interesting use of UAVs. See the following section for specific comment.

Specific Comments

Line 5: Change “In our study we” to “In our study, we”

Line 10: Change “For the sub-meter change analysis we” to “For the sub-meter change analysis, we”

Line 14: Change “In a defined point cloud subset of an erosion feature we” to “In a defined point cloud subset of an erosion feature, we”

Line 15-16: “Projecting the development of the erosion feature we” to “Projecting the development of the erosion feature, we”

Lines 18-19: “For a future improvement of the workflow we” to “For a future improvement of the workflow, we”

Line 42: Delete “for example”.

Line 84: Change “Based on ArcticDEM data [21] the” to “Based on ArcticDEM data [21], the”

Line 86: Change “In the center of it the” to “In the center of it, the”

Line 87: Delete “pull out”

Line 88: Change “Parallel to the highway on the eastern side a trench like” to “Parallel to the highway on the eastern side, a trench-like”

Line 90: Delete “highly”

Line 92: Change “plays” to “is”

Line 99: Change “To obtain images in nadir we” to “To obtain nadir images, we”

Line 100: Delete “flying”

Line 101: Change “80 m” to “80 m above ground level (a.g.l.)” and change “Additionally we” to “Additionally, we”

Line 102-103: Change “60 m with” to “60 m a.g.l. and”

Line 103-104: Make final sentence part of preceding paragraph and change “In this acquisition mode the” to “In this acquisition mode, the”

Line 124: Delete “us with”

Lines 128-129: Change “with parallel flight paths and in nadir only,” to Change “with parallel, nadir flight paths”

Figure 3 Caption: Change “Acquisitions were collected in nadir direction at a height of 80 m,” to “Acquisitions were collected at nadir and height of 80 m a.g.l.,” and change “the flight path of the point of interest flight at a height of 60 m” to “the flight path covering points of interest and at a height of 60 m a.g.l.” and change “(RGB) as processed” to “(RGB) processed”

Line 135: Change “of 60m” to “of 60m a.g.l.”

Line 139: Change “Subsequently the” to “Subsequently, the”

Line 181: Change “For all point cloud related processing steps we” to “For all point cloud processing steps, we”

Line 184: Change “with a high detail of information” to “with detailed information”

Line 224: Change “optimising” to “optimizing”

Line 261: Change “multi-scale model-to-model” to “multi-scale, model-to-model”

Line 281: Change “ the M3C2 distance we” to “ the M3C2 distance, we”

Line 287: Change “analyse” (British English spelling) to “analyze”(American English spelling)

Line 288-289: Change “we rasterized the attribute of the vertical displacement (Dz) of the point cloud by using” to “we generated a raster of the vertical displacement (Dz) of the point cloud using”. This suggested change avoids the confusion between British English (rasterise) and American English (rasterize) for and international audience.

Line 291: Change “metres per pixel” to “m per pixel” to avoid confusion for an international audience.

Line 298: Change “We chose a categorization into two classes” to “We chose two classes”

Line 300: Change “,which we subsequently vectorized to mask” to “and constructed a vector overlay to mask”

Line 341: Change “analysing” to “examining” to avoid confusion for an international audience.

Line 397: Change “In rasterising the point cloud to” to “The raster point cloud with a” and change

“we found that a pixel size of 5 x 5 m” to “had a pixel size of 5 x 5 m that”

Line 403: Change “ we identified the detected changes as changes in water level” to “we detected changes in water level”

Line 412: Change “Moreover we” to “Moreover, we”

Line 413: Change “persisting” to “consistent”

Line 419: Change “persisting” to “consistent”

Line 420: Change “trench like” to trench-like

Line 421: Change “At the southern edge we” to “At the southern edge, we”

Line 424: Change “as artifacts” to “artifacts”

Figure 7b is a good summary of the technique and results.

Line 437: How does a dimension of 20 pixels relate to meters in the x and y dimension? And how do these dimension relate to the errors of the technique? Is the error 0.2 m with the level IV processing?

Line 449: Change “pooling” to “standing”

Line 458: Change “with kNN4 we” to “with kNN4, we”

Line 460: Change  ”section 2.3.1) we” to ”section 2.3.1), we”

Line 471: Change “persisting” to “consistent”

Line 478: Change “UAV.Our” to “UAV.  Our”

Line 479: Delete “for use”. It is not necessary.

Line 482: Change “rasterization” to “raster”

Line 487: Change “persisting” to “consistent”

Line 491: Change “-0.35 we” to “-0.35, we”

Line 494: Change “Yet it” to “However, it”

Line 496: Suggested change- “Analysing” to “Assessing”

Line 499: Change “ice wedges of the polygons” to “ice wedge polygonal network”

Line 502: Change “analysed” to “performed”

Line 506 and 507: Change “North and Southwest” to “north and southwest” and change “During the 2018 image acquisition we” to “During the 2018 image acquisition, we”

Line 509: Delete “process” and delete “we showed”

Line 521: Delete “such as ours”

Line 522: Change “scenarios we” to “scenarios, we”

Lines 530-531: Change “can further recommend to implement” to “recommend”

Line 533: Change “For an advancement of the cloud comparison” to “For improving cloud comparisons” and change “are characterized by” to “have”

Line 545:  Delete “for example”. It is not necessary.

Line 549: Change “fast consumer-grade” to “fast, consumer-grade”

Lines 551-555: I an not convinced that these statements are supported by the results presented. It would seem that several years (3-5 years) of data collection using this technique might be more useful for the interested citizen/ environmental volunteer.

Line 555: By “larger scale” do you mean a larger spatial extent? If you are referring to a larger spatial extent, then it is small-scale. The term “small scale” covers a large area and the term “large scale” covers a detailed or small area.

Author Response

This manuscript is a resubmission of an earlier submission. The following is a list of the peer review reports and author responses from that submission.

Round 1

Reviewer 1 Report

Dear authors, I have read very carefully the work entitled "The Potential of UAV Imagery for the Detection of Rapid Permafrost Degradation: Assessing the Impacts on Critical Arctic Infrastructure" and I consider that it addresses a subject that needs to be studied. The presence of permafrost and its degradation due to temperature increases is a situation that will be with us for a long time. The consequences of permafrost degradation are many and some can be very harmful to people. Therefore, I consider that a work focused on finding the most efficient and fastest way to detect these changes in surface morphology is very valuable. I also consider it very important that the objective of the work is to find a methodology that is easy to use for any citizen. Problems due to permafrost degradation are going to be widely distributed and people, as well as the scientific community and governments, will need tools for their analysis.

On the other hand, I consider that although the central theme of the work is the analysis of the workflow to detect changes in the surface, there should be a slightly more detailed background on the type and distribution of permafrost in that region. The erosive and subsidence processes that are occurring in that sector should also be given a little more detail.

Minnor comments have been made in the attached pdf file.

Reviewer 2 Report

Overall, the paper is innovative and provides a methodology that is open and free to users. This allows for increased data for large-scale analysis, while also empowering citizen scientists to contribute to understanding of our world. I’d like to see this paper get published, following resolution of my comments below: 

Overarching issue: There is a lack of a validated procedure. It is difficult to tell how accurate this process is without any comparison to an accepted methodology. There are numerous articles published that use UAV-derived orthomosaics and DSMs, using provided software (e.g., Pix4D). It would be helpful to run this procedure through proven software to compare results on offset. Were there GPS data taken during surveying? For validation as opposed to part of the proposed methodology.

Nomenclature (P. ← page, L. line(s))

P.6 L.89-92: This was confusing. The 2.2 link pushed to the previous page but I can't find an easily accessible explanation of this process. Since source 23 is behind a paywall, I would like to see a small paragraph that provides a general review of this procedure. 

P.11 L.252-264 you describe RMS and offsets. I am still confused as to how you quantified the offset – relative to what? Did you use other tools to verify position or coordinates? 

P.17 L. 336-338: This statement has significant implications but there is very little explanation of what work you did and how you reached this modeling conclusion. This needs to be elaborated and explained thoroughly. What evidence do you have that suggests similar behavior between 2020 and 2040? 

Reviewer 3 Report

In this study the authors monitor ground subsidence near Dalton Highway, Alaska, based on UAV point clouds acquired near the same time period for two years, and predict when future impacts on the highway are likely to occur. The point cloud bias was improved by algorithm improvements. It is an interesting attempt in the circumpolar Arctic region and could provide an example for low-cost permafrost monitoring in small areas. I recommend to accept it for publication after major revisions. I list my major and minor concerns as follows:

Major concerns

1.      I’m confused about how to use one year UAV-based displacement alone to evaluate the development of erosion features in the coming twenty years. The development of erosion may depend on local subsurface geomorphological and geological conditions.

2.      Table 1: Are the modifications to WebODM Lightning based on the author's experience or do they have specific references? This is a critical step that affects the quality of the point cloud.

3.      Line 156: Whether do the lack of ground control points have a significant impact on experimental results is a key question? Please clarify this point.

4.      The authors used UAV technique to map ground deformation in fine scale. Despite Figure 4 shows the locations suffering from high vertical displacement, I think a quantitative deformation map is necessary.

5.      Figure 7: Why do you resample vertical displacements from centimetre spatial resolution to 10m x 10m? Is it only used to reduce the pretzel noise of the results?

Minor concerns

Line 21-22: The keywords could be more focused. “permafrost” and “degradation” can be combined and “Dalton Highway” can be deleted.

Figure 1 and 3: It is suggested that the red band images be replaced with RGB images of the study area, which would present the reader with more information about the study area.

Line 80: Is “visible spectrum” RGB? Please specify the specific spectral band information obtained.

Figure 2: The structure of the figure can be optimized, and the current version takes up a lot of space but offers less information. In addition, the specific processing instructions should not appear in the figure name, please move them to the text and add the appropriate reference.

Line 107-120: Are these steps given in the software’s operating documentation? If so, it is recommended to give references to the operating documentation or to show them in a concise flow chart.

Reviewer 4 Report

General Comments

 The authors conducted an interesting study using an off-the-shelf UAV and open-source software. In general, the manuscript is well-written and illustrated. Minor changes can improve an otherwise clearly written manuscript. See specific comments.

Specific Comments

 Lines 4 and 10: Change “small-scale” to “large-scale” here and elsewhere in the document. Small-scale covers a large spatial extent whereas large scale covers a small spatial extent

Line 149: Change “Since” to “Because”. The word “because” infers a reason whereas “since” infers a time.

Line 182: Change “Therefore” to “Therefore,”.

Line 230-231: The very coarse resolution of 10 m spatial resolution and subsequent Sobel filter to highlight edges seems counterproductive given the initial image acquisition spatial resolution.

Line 274: Change “joint” to “point”.

Line 307: How would the results change if the spatial resolution was 5 m?

Figures 5 and 6 are useful for the reader.

Line 328: Change “artefacts” to “artifacts”. It is clear that British spelling is used through out the manuscript. Is “artefact” a British spelling?

Figure 7 shows a nice network of polygons east and west of the stable junction. It would be helpful to point them out. These polygons appear easier to see than in Figure 1. Also, most areas of high vertical displacement appear along the river bank and not the polygonal network.

Figure 8.  It would be nice to have this figure showing the same area as Figure 7 and Figure 1.

Line 345: Change “lense” to “lens”.

Line 357: It’s good to estimate the offset at image edges due to lack of tie points.

Line 369: Change “artefacts” to “artifacts”. It is clear that British spelling is used through out the manuscript. Is “artefact” a British spelling?

Lines 372-374: It is good to acknowledge these limitations.

Line 382: Change “Since” to “Because”. The word “because” infers a reason whereas “since” infers a time.

Lines 398-403: Agreed!

Line 406 and 410: Change “small-scale” to “large-scale” here and elsewhere in the document. Small-scale covers a large spatial extent whereas large scale covers a small spatial extent.
